# Cervical Cancer Prevention Knowledge (Cckp-64) among Female Students in Novi Sad, Serbia during COVID-19 Pandemic

**DOI:** 10.3390/healthcare11101400

**Published:** 2023-05-12

**Authors:** Sanja D Tomić, Andrijana Ćorić, Slobodan Tomić, Ermina Mujičić, Jelena Malenković, Armin Šljivo, Goran Malenković

**Affiliations:** 1Faculty of Medicine, University of Novi Sad, 21000 Novi Sad, Serbia; 2Clinical Center, University of Sarajevo, 71000 Sarajevo, Bosnia and Herzegovina

**Keywords:** women’s health, genital neoplasms, female, genital diseases, female, Serbia, disease

## Abstract

Cervical cancer is a significant global health concern affecting young women, with over 500,000 new cases reported annually. This questionnaire-based study aimed to evaluate the knowledge of cervical cancer prevention among female students at the University of Novi Sad during the COVID-19 pandemic using the Cervical Cancer Knowledge Prevention-64 (CCKP-64) tool. The study sample consisted of 402 predominantly 20–22-year-old female students from either social or technical science faculties in urban environments. Results revealed that out of the 402 female students involved in the study, most had a good general knowledge of primary prevention of cervical cancer, with a correct answer rate ranging from 29.9 to 80.6%. On the contrary, only 63.4% of female students have heard about the vaccine against cervical cancer; 52.0% know that the vaccine exists in Serbia; and 31.8% know where to get vaccinated. Only a small proportion of students (9.7%) have encountered cervical cancer among their relatives/friends and think that the disease could affect them in the future (25.4%). Older students (>26 years) generally (*p* < 0.05) had better knowledge regarding distressing symptoms of cervical cancer, cytological examination and secondary prevention; however, it was also noted that a significant percentage of this age group reported not having received vaccinations (53.0%, *p* = 0.001). This study underscores the need for increased awareness and education about the HPV vaccine and secondary prevention among young women in Serbia. Future research should investigate knowledge and attitudes toward cervical cancer prevention in diverse populations to develop effective interventions and strategies. These findings have implications for public health policies in Serbia to promote cervical cancer prevention among young women.

## 1. Introduction

Cervical cancer is currently the second most common cancer among young women aged 15 to 44, with over 500,000 new cases reported each year globally [1]. It ranks as the third most common cancer type [2], with approximately 200,000 deaths per year worldwide [3] and the highest rates of mortality and incidence in developing sub-Saharan countries. Two-thirds of all cervical cancers are diagnosed in women between the ages of 35 and 66, with a median age of 49 [4]. While developed countries have a rather good prognosis for cervical cancer, developing countries account for 87% of all cervical cancer-related deaths [3]. Cervical cancer is a preventable disease, and both screening with Pap tests and vaccination against the human papillomavirus (HPV) are important strategies for its prevention. When detected early through regular Pap smears, the five-year survival rate for cervical cancer can be as high as 92% [5].

The HPV vaccine has been shown to be highly effective in preventing the types of HPV that are most commonly associated with cervical cancer by up to 90% [6]. Despite the effectiveness of Pap tests and HPV vaccination, many women are not receiving these preventive measures. According to a recent study, only 68% of eligible women in the United States receive regular Pap tests, and only 54% of adolescent girls receive the recommended HPV vaccine [7]. This highlights the need for increased education and access to preventive measures for cervical cancer.

A study conducted among female university students in Serbia found that the prevalence of HPV infection, the primary cause of cervical cancer, was alarmingly high. The study revealed that over 60% of the participants were infected with at least one type of HPV [8]. The high prevalence of HPV infection among university students in Serbia can be attributed to various factors, including poor sexual health education, a lack of access to HPV vaccination and insufficient screening programs. Furthermore, studies have shown that young women in Serbia have inadequate knowledge about cervical cancer and its prevention. This is reflected in their low participation rates in cervical cancer screening programs. Cervical cancer screening is essential to detect pre-cancerous changes early and prevent their progression to cervical cancer. The lack of awareness and education regarding cervical cancer and screening among female university students in Serbia is a cause for concern, as it can lead to missed opportunities for early detection and treatment [9]. Efforts have been made to improve cervical cancer prevention and control in Serbia, including the implementation of a national HPV vaccination program in 2016 [10]. However, there is still a need for increased education and awareness among university students and the general population about cervical cancer and its prevention. It is essential to promote sexual health education, provide accessible and affordable screening programs, and increase awareness about the HPV vaccine.

Although official data on cancer cases diagnosed during the COVID-19 pandemic has not yet been published, we have noticed a significant reduction in the number of newly diagnosed cervical cancer cases and women seeking cervical cancer testing in our clinical practice during the COVID-19 pandemic. In response to the COVID-19 pandemic, healthcare facilities have implemented strategies to enhance their ability to accommodate patients with COVID-19 while concurrently minimizing the movement and flow of non-COVID-19 patients. Medical institutions have been reducing outpatient visits and postponing procedures, such as elective surgery, testing and treatments, which may disproportionately affect cancer patients since the interval between diagnosis and the start of the treatment may have a detrimental impact on outcomes. By advising people to postpone non-emergency consultations, tests and treatments while under quarantine, national screening programs were temporarily suspended in certain places to reduce the burden on health services [11].

This questionnaire-based study aimed to evaluate the knowledge of cervical cancer prevention among female students at the University of Novi Sad during the COVID-19 pandemic using the Cervical Cancer Knowledge Prevention-64 (CCKP-64) tool. The pandemic has disrupted healthcare services, including cervical cancer screening and vaccination programs, making it crucial to assess the level of knowledge about cervical cancer prevention among the population. The results of this survey will give us a quick overview of the current level of understanding about cervical cancer prevention, help us stay informed about the current state of affairs, and also draw the attention of government bodies to the importance of educating female students about preventing cervical cancer. This research makes a significant contribution to the literature in the field of cervical cancer prevention, particularly during the COVID-19 pandemic. By investigating the level of knowledge about cervical cancer prevention among students, our study provides insight into the effectiveness of current educational programs on cervical cancer prevention and identifies areas for improvement. Our findings can inform the development of new educational strategies and interventions to increase awareness and understanding of cervical cancer prevention.

## 2. Materials and Methods

### 2.1. Subjects

This cross-sectional study was performed during the academic year 2021–2022, among the students at the University of Novi Sad in Serbia, during the COVID-19 pandemic. The study was approved by the Ethical Committee of the University of Novi Sad (01-39/202/1) with the exception of the field of medical sciences. Students completed an anonymous online questionnaire regarding Cervical Cancer Knowledge Prevention-64 (CCKP-64) using the Google Forms Administration App, which prevented multiple responses per email. The collection of official email addresses assigned to students by the University of Novi Sad was achieved through their information system, while the selection of participants for the study was carried out using convenience sampling. The minimum sample size computed for the student population of the University of Novi Sad using $n^{\prime }=\frac{n}{1+\frac{z2xp(1-p)}{e2N}}$ (*z*—*z* score; *e*—margin of error; *N*—population size; *p* population proportion) was 355 students (*z* = 355, *e* = 5%, *N* = 4514). The inclusion criteria were (i) female students of the University of Novi Sad, (ii) at least 18 years old and (iii) having not been diagnosed with cervical cancer or (iv) having undergone a hysterectomy. The criteria for excluding participants from the study were (i) male individuals, (ii) female students studying at the Faculty of Medical Sciences, (iii) not studying at the University of Novi Sad, (iv) those under 18 years of age, or (v) those who did not give their consent to participate in the study. Before the study began, all participating students were given information about the study’s purpose, the data that would be collected and used, the anonymity of the data, and how to complete the questionnaire. They were also provided with online written consent forms. All aspects of the study were carried out in compliance with the Helsinki Declaration. 

### 2.2. Data Collection and Study Questionnaire

Developed based on the CCKP-64 [12], the questionnaire, which consisted of 64 close-ended questions and 67 variables distributed across six domains and was based on adapted guidelines from the European Organization for Research and Treatment of Cancer (EORTC), is designed to assess knowledge of cervical cancer prevention among high school and college female students. 

The first domain pertains to the sociodemographic characteristics of the respondents, such as age, education and place of origin. The second domain examines general knowledge about cervical cancer prevention in the context of 6 questions. The third domain is intended to explore the relationship between the assessed risk factors and the occurrence of the disease. Within this domain, 17 factors are listed, which the respondent marks on a six-point Likert scale, where 0 indicates no connection between the listed factor and the occurrence of the disease and 5 indicates a significant connection. The fourth domain represents an assessment of knowledge about primary cervical cancer prevention. It is divided into three sub-domains: A—lifestyle, B—vaccine and C—the best age for HPV vaccination. The fifth domain is intended to assess knowledge about secondary cervical cancer prevention. It is divided into two sub-domains, where domain A pertains to the relationship of the listed symptoms with cervical cancer and domain B pertains to knowledge about the Pap test. The sixth domain explores the sources of information through which the respondents obtained information about cervical cancer. Cronbach’s alpha coefficient of the CSS was 0.914 (n = 64), indicating a very high level of reliability [13].

### 2.3. Statistical Analysis

The study analyzed data on knowledge of cervical cancer prevention using descriptive statistics. The Statistical Package for Social Sciences (SPSS) in IBM Statistics v26.0 was used to analyze the collected data. If the data had a normal distribution, they were presented with a mean ± SD, and if not, they were presented with a median (25th or 75th percentile). An independent samples *t*-test was used to compare continuous variables between groups, and a one-way analysis of variance (ANOVA) was used for parametric data. Chi-squared tests were used to identify differences in categorical variables between groups where appropriate. Cronbach’s alpha coefficient was used to evaluate the internal consistency of the scales used in the study. A *p*-value of less than 0.05 was considered statistically significant.

## 3. Results

In total, 402 students were included in the study from a population of *N* = 4514, with an estimated participation rate of 8.9%. Our study sample was predominantly in the age group of 20 to 22 years old (158, 39.3%), from either social science (92, 22.9%) or technical science (101, 25.1%), and from urban environments with between 10,000 and 100,000 inhabitants (214, 53.2%). All other demographic characteristics of the study sample are presented in Table 1.

### 3.1. General Knowledge Regarding Cervical Cancer among Female Students at the University of Novi Sad

Out of 402 students engaged in the study, 363 (90.3%) have heard about cervical cancer, and 272 (67.7%) agree that cervical cancer can be a terminal illness. Only 198 students (49.3%) agree that cervical cancer can be associated with an infection, while even fewer students (146, 36.3%) agree that there is an effective method that significantly reduces the risk of the disease. Only a small proportion of students (39, 9.7%) have encountered cervical cancer among their relatives/friends, and 102 students (25.4%) think that the disease could affect them in the future. All other data regarding general knowledge of cervical cancer among female students of the University of Novi Sad are presented in Table 2.

### 3.2. Relationship between Estimated Risk Factors and Occurrence of the Disease Cancer among Female Students of the University of Novi Sad

Risk factors for cervical cancer were evaluated using a Likert scale ranging from zero to five, where zero indicated no correlation and five indicated a very strong correlation between the risk factor and the occurrence of the disease. A decent proportion of participants (40%) rated the relationship between genetic factors and the occurrence of cervical cancer as four or five, while 48.2% rated the relationship between HPV infection and the occurrence of cervical cancer in the same way. The relationship between estimated risk factors and the occurrence of the disease is presented in Table 3.

Among participants of different age groups, no statistically significant difference was found in the total score of questions about the correlation between risk factors and the occurrence of cervical cancer (M ± SD = 46.56 ± 9.337, 48.76 ± 9.227, 49.34 ± 9.617, 48.83 ± 10.653, *p* = 0.327).

### 3.3. Knowledge about Primary Prevention of Cancer among Female Students of the University of Novi Sad

The results regarding knowledge of primary prevention of cervical cancer showed that the majority of female students were relatively aware of the correlation between proper lifestyle habits and the onset of cervical cancer, with rates of correct answers ranging from 29.9 to 80.6%. On the contrary, only 63.4% of female students have heard about the vaccine against cervical cancer; 52.0% know that the vaccine exists in Serbia; and 31.8% know where to get vaccinated. The results related to knowledge of primary prevention of cervical cancer are presented in Table 4. Regarding questions about lifestyle as a measure of primary prevention, there was no statistically significant difference in responses among participants of different age groups. However, when asked “Have you heard of a vaccine against cervical cancer?”, participants aged 20–22 more frequently answered “no” compared to the other groups (*p* = 0.001). Significantly more subjects over 26 years old responded that they had not been vaccinated (53.0%, *p* = 0.001). For the questions “Is the vaccine free?” and “What is the best age for vaccination?”, subjects over 26 years old gave significantly more correct answers (*p* = 0.002, *p* = 0.001). Regarding the question “Is the vaccine available in Serbia?”, subjects between the ages of 23 and 25 gave significantly more correct answers (*p* = 0.008). There were no statistically significant differences among the responses of subjects of different ages to the remaining vaccination questions. The differences in exact answers regarding primary prevention of cervical cancer are ordered among subjects of different age categories and presented in Table 5.

### 3.4. Knowledge about Secondary Prevention of Cancer among Female Students of the University of Novi Sad

The results regarding knowledge of secondary prevention of cancer among female students showed relatively poor knowledge regarding distressing symptoms and cytological examination. The results regarding secondary prevention of cervical cancer are presented in Table 6. For most questions about recognizing alarming symptoms as a measure of secondary prevention, participants over 26 years old gave statistically significantly more accurate answers (“Heavy menstruation or bleeding between periods” *p* = 0.001, “Irregular menstruation or absence of menstruation” 59.1%, *p* = 0.004, “Appearance of vaginal discharge with an unpleasant odor” *p* = 0.000, “Appearance of mucus-like, bloody discharge” *p* = 0.000, “Appearance of bleeding after sexual intercourse” *p* = 0.000, “High temperature” *p* = 0.000). For the remaining three questions about alarming symptoms, participants from different age groups did not give statistically significantly different answers. Regarding the question “Have you ever heard of a Pap test?”, participants over 26 years old answered “yes” more frequently than others (98.5%, *p* = 0.003), while participants aged 18–19 answered less frequently than others (69.5%, *p* = 0.001). For most questions about Pap testing, participants from different age groups did not give statistically significantly different answers. 

A statistically significant difference in answers existed for three questions (“Does the Pap test provide a 100% chance of early detection of cervical cancer?”, “Is free testing available?”, “How often should women do the Pap test?”) where participants over 26 years old gave less accurate answers to the first question and more accurate answers to the remaining two questions (*p* = 0.000, *p* = 0.000, *p* = 0.000). Regarding the question “Do you think you should do the Pap test?”, women over 26 years old were statistically significantly more likely to answer “yes” (*p* = 0.000), while the two youngest age groups (18–19, 20–22) were statistically significantly more likely to answer “no” (*p* = 0.011, *p* = 0.000).

Differences in correct answers about secondary prevention between participants belonging to different age categories are presented in Table 7. For the question “How often should women do a Pap test?”, girls older than 26 years of age significantly more often gave a correct answer (*p* = 0.000), while the youngest age category (18–19) significantly less often gave a correct answer (*p* = 0.002).

## 4. Discussion

To our current understanding, this investigation represents one of the initial inquiries carried out in the West Balkan region to explore the level of knowledge related to cervical cancer prevention among female students amidst the COVID-19 pandemic. Our study sample, which was mostly 20 to 22 years old, from either social science or technical science faculty and from urban environments, had good general knowledge regarding cervical cancer and was relatively aware of the correlation between proper lifestyle habits and the onset of cervical cancer. On the contrary, a significant proportion of female students have not heard about the vaccine against cervical cancer and do not know that the vaccine exists in Serbia. Significantly more subjects over 26 years old responded that they had not been vaccinated. The results regarding knowledge of secondary prevention of cancer among female students showed relatively poor knowledge regarding distressing symptoms and cytological examination. Older students showed significantly better knowledge regarding distressing symptoms of cervical cancer, cytological examination and secondary prevention.

Our study’s findings on the level of knowledge related to cervical cancer prevention among female students in the West Balkan region are consistent with previous research conducted in Serbia [14,15]. Rančić et al. [14] and Kesic et al. [15] reported that young women generally have poor knowledge and negative attitudes toward cervical cancer prevention. These findings suggest that educational programs aimed at increasing awareness of cervical cancer and its prevention may have had some success in Serbia. However, our study also found that a significant proportion of female students had not heard about the vaccine against cervical cancer and were not aware of its availability in Serbia. This is a concerning finding, as HPV vaccination is an essential component of cervical cancer prevention, especially during the COVID-19 pandemic [16]. Sitaresmi et al. [17] and also Marić et al. [18] highlight the need for increased awareness and education about HPV vaccination among both young women and their parents. These findings suggest that there is still a need for more extensive educational interventions to increase awareness of cervical cancer and its prevention, especially with regard to the availability and benefits of the HPV vaccine. Additionally, it is important to address concerns and criticisms about HPV vaccination in order to promote its uptake and effectiveness. While some studies have raised concerns about the safety and long-term effectiveness of the vaccine, the vast majority of research has supported its safety and efficacy [19]. Despite the effectiveness of HPV vaccination, vaccination rates vary considerably among different countries and regions, and some individuals may be hesitant to receive the vaccine due to concerns about safety or a perceived lack of necessity, even though recent studies have shown that the vaccine has resulted in a significant reduction in the prevalence of HPV infections and related diseases, including cervical cancer [20]. Vaccine hesitancy may arise due to various factors, including concerns about vaccine safety, a perceived lack of necessity and misinformation or mistrust of health authorities [21]. A recent systematic review of studies on HPV vaccine hesitancy identified several common reasons for vaccine hesitancy, including concerns about vaccine safety, a lack of knowledge about HPV and its associated diseases, and a perceived low risk of HPV infection [22]. To address vaccine hesitancy and improve HPV vaccination uptake, several strategies have been proposed. One approach is to improve vaccine education and communication, including providing accurate information about vaccine safety and efficacy, addressing common myths and misconceptions, and engaging with communities and stakeholders to build trust and increase vaccine acceptance [23]. Another approach is to implement vaccine promotion and outreach campaigns, such as school-based vaccination programs, reminder systems for healthcare providers and patients, and incentives for vaccination [24]. Even though the government has accepted HPV vaccination as obligatory [25], there is still some hesitancy among young females, which could be related to poor knowledge regarding cervical cancer prevention.

Furthermore, our study found that older students generally showed better knowledge regarding distressing symptoms of cervical cancer, cytological examination and secondary prevention, which could be attributed to their more advanced academic training and exposure to relevant coursework or practical experience compared to younger students who may still be in the early stages of their education. Additionally, older students may have had more time and opportunities to seek out information on cervical cancer and related topics, or they may have personally known someone who has been affected by the disease. Nonetheless, further investigation is necessary to confirm these potential explanations and determine the most effective strategies for educating all students on this important health issue.

Our study is subject to several limitations that must be acknowledged. First, due to the cross-sectional design of our study, it is difficult to establish causality. Secondly, our sample only included students from the University of Novi Sad, which limits the generalizability of our findings to other regions of the country. To improve the generalizability of future studies, it is important to include a more diverse sample from various regions. Thirdly, our study focused primarily on students from urban environments, which may not accurately reflect the experiences of patients living in rural or remote areas. Therefore, future studies should aim to include a broader representation of students, particularly those from vulnerable populations residing in remote areas. Future studies should also include students from diverse majors, a larger sample and possibly different universities in Serbia to calculate the differences between regions in the country.

## 5. Conclusions

In conclusion, our study found that female students in Serbia during the COVID-19 pandemic generally have a good level of knowledge regarding cervical cancer prevention, but there is still room for improvement in terms of awareness and education about the HPV vaccine and secondary prevention. Older students showed better knowledge about distressing symptoms of cervical cancer, cytological examination and secondary prevention, highlighting the need for targeted interventions among younger populations. While there have been criticisms and concerns about HPV vaccination, the overwhelming evidence supports its safety and effectiveness in preventing cervical cancer. Therefore, promoting awareness and education about cervical cancer prevention and HPV vaccination among young women and their parents is crucial to reducing the burden of cervical cancer in Serbia and globally. Future research should continue to investigate the knowledge and attitudes of different populations toward cervical cancer prevention in order to develop effective interventions and strategies.

## Figures and Tables

**Table 1 healthcare-11-01400-t001:** Age, field of study and living environment of the female students of the University of Novi Sad during the COVID-19 pandemic.

Variable	Percentage	Frequency
Age		
17–19	14.7%	59
20–22	39.3%	158
23–25	29.6%	119
>26	16.4%	66
Field of study		
Natural sciences	14.2%	57
Technical sciences	25.1%	101
Humanistic sciences	17.4%	70
Social sciences	22.9%	92
Art sciences	6.0%	24
Other	14.4%	58
Living environment		
Rural environment	6.2%	25
Urban environment < 10,000 inhabitants	23.1%	93
Urban environment 10,000–100,000 inhabitants	53.2%	214
Urban environment > 100,000 inhabitants	17.4%	70

**Table 2 healthcare-11-01400-t002:** General knowledge regarding cervical cancer among female students at the University of Novi Sad.

Questions	Yes	No	I Do Not Know
	% (*N*)	% (*N*)	% (*N*)
Have you ever heard of cervical cancer?	90.3 (363)	9.5 (38)	0.2 (1)
Can cervical cancer be a terminal illness (or can you die from cervical cancer)?	67.7 (272)	3.5 (14)	28.9 (116)
Can cervical cancer be associated with an infection?	49.3 (198)	4.7 (19)	46 (185)
Is there an effective method that significantly reduces the risk of this disease?	36.3 (146)	28.9 (116)	34.8 (140)
Have you ever had direct contact with the disease (e.g., has any of your relatives or friends suffered from it)?	9.7 (39)	85.6 (344)	4.7 (19)
Do you think this disease could affect you in the future?	25.4 (102)	15.9 (64)	58.7 (236)

**Table 3 healthcare-11-01400-t003:** General knowledge regarding cervical cancer among female students of the University of Novi Sad.

Factors	M	SD	Factor	M	SD
Young age	0.96	1.320	Miscarriages and abortions	2.54	1.596
Genetic factor	2.97	1.544	A large number of pregnancies and childbirths	2.38	1.620
HPV infection	3.32	1.494	Early menarche	1.11	1.490
HIV infection	3.21	1.444	Use of condoms	1.20	1.512
Multiple sexual partners	3.48	1.507	Hormonal contraception	1.87	1.573
Early sexual initiation	3.09	1.702	Breast feeding	1.25	1.462
History of STDs	3.58	1.243	Use of drugs or psychoactive substances	2.12	1.629
Alcohol abuse	2.05	1.585	Using public swimming pools	1.95	1.569
Smoking	2.27	1.593			

M—mean, SD—standard deviation.

**Table 4 healthcare-11-01400-t004:** Primary prevention knowledge regarding cervical cancer among female students at the University of Novi Sad.

A. Lifestyle				B. Vaccine			
Question	Yes	No	I Do Not Know	Question	Yes	No	I Do Not Know
A diet rich in ‘so-called’ antioxidants?	70.1%	8.5%	21.4%	Have you heard about the vaccine ‘against cervical cancer’?	63.4%	32.1%	4.5%
Regular physical exercise?	78.6%	11.4%	10.0%	If such a vaccine exists, is it available in Serbia?	52.0%	6.7%	41.3%
Use of vitamin supplements?	62.9%	16.4%	20.06%	Is it free of charge (reimbursed by the National Health Fund)?	19.2%	29.6%	51.2%
Proper, long and relaxing sleep?	80.6%	8.2%	11.2%	Does it guarantee 100% protection from cervical cancer?	4.2%	54.2%	41.5%
Avoiding highly processed food?	60.7%	16.9%	22.4%	Do you know where you can get vaccinated?	31.8%	28.9%	39.3%
Avoiding genetically modified food?	67.2%	11.4%	21.4%	Have you ever been vaccinated?	61.9%	32.8%	5.2%
Weight loss?	29.9%	29.6%	40.5%				
Restraining from casual sex?	60.7%	16.4%	22.9%				
C. What is the best age (year) to get vaccinated?					8 years		20.1%
					9–13 years		24.9%
					14–18 years		26.6%
					19–25 years		21.1%
					>25 years		7.2%

**Table 5 healthcare-11-01400-t005:** Primary prevention knowledge regarding cervical cancer among female students of the University of Novi Sad in comparison to different age groups.

Questions	Age Groups				χ^2^	*p*
	18–19	20–22	23–25	>26		
A diet rich in ‘so-called’ antioxidants?	40 (67.8%)	109 (69.0%)	85 (71.4%)	48 (72.7%)	2.314	0.889
Regular physical exercise?	49 (83.1%)	126 (79.7%)	89 (74.8%)	52 (78.8%)	8.069	0.233
Use of vitamin supplements?	36 (61.0%)	98 (62.0%)	81 (68.1%)	38 (57.6%)	2.543	0.864
Proper, long and relaxing sleep?	45 (76.3%)	129 (81.6%)	99 (83.2%)	51 (77.3%)	1.898	0.929
Avoiding highly processed food?	37 (62.7%)	96 (60.8%)	67 (56.3%)	44 (66.7%)	3.634	0.726
Avoiding genetically modified food?	38 (64.4%)	106 (67.1%)	74 (62.2%)	52 (78.8%)	7.545	0.273
Weight loss?	17 (28.8%)	45 (28.5%)	40 (33.6%)	18 (27.3%)	5.949	0.429
Restraining casual sex?	38 (64.4%)	97 (61.4%)	71 (59.7%)	38 (57.9%)	8.860	0.182
If such a vaccine exists, is it available in Serbia?	24 (40.7%)	69 (43.7%)	76 (63.9%)	40 (60.6%)	23.203	0.001 *
Is the vaccine safe?	11 (18.6%)	23 (14.6%)	20 (16.8%)	23 (34.8%)	20.831	0.002 *
Does it guarantee 100% protection from cervical cancer?	32 (54.2%)	78 (49.4%)	73 (61.3%)	35 (53.0%)	8.103	0.231
What is the best age (year) to get vaccinated?	10 (16.9%)	37 (23.4%)	25 (21.0%)	28 (42.4%)	34.670	0.001 *

* values are significant.

**Table 6 healthcare-11-01400-t006:** Secondary prevention knowledge regarding cervical cancer among female students of the University of Novi Sad.

A. Distressing Symptoms—Select the Symptoms which May be Associated with the Presence of Cancer			B. Cytological Examination		
Question	Yes	No	Question	Yes	No
Lack of symptoms from genital areas?	25.9%	74.1%	Have you ever heard about cytological examination?	84.8%	15.2%
Painful menstruation	32.3%	67.7%	Is it a test that gives a 100% chance of early diagnosis of cervical cancer?	36.8%	63.2%
Intensive periods or bleeding between periods?	47.8%	52.2%	Is the test painful?	21.9%	78.1%
Irregular menstruation or lack of menstruation?	40.8%	59.2%	Is it a time-consuming test?	5.2%	94.8%
Smelly vaginal discharge?	45.5%	54.5%	Is it possible to be tested free of charge?	47.5%	52.5%
Blood-stained mucus?	62.2%	37.8%	Is it sufficient to do the test only once in order to eliminate the risk of cervical cancer?	10.0%	90.0%
Itching in the genital area?	32.8%	67.2%	Can the test cause serious complications?	5.7%	94.3%
Bleeding after intercourse?	44.0%	56.0%	Is it possible for the Pap smear to increase the susceptibility to cervical cancer in the future?	4.7%	95.3%
High fever?	40.8%	59.2%	Do you think you should undergo cytological examination?	60.0%	40.0%
			At what age (year) can women in Serbia undergo cytological examination free of charge?	17–25 years	73.9%
				26–59 years	11.2%
				>60 years	14.9%
			How long (year) after sexual initiation should women undergo the test?	<1 years	66.2%
				1–3 years	32.8%
				3–6 years	0.5%
				>6 years	0.5%
			How often (year) should women do the test?	Every year	71.6%
				Every 3 years	25.9%
				Every 10 years	2.2%
				Only once	0.2%

**Table 7 healthcare-11-01400-t007:** Secondary prevention knowledge regarding cervical cancer among female students of the University of Novi Sad in comparison to different age groups.

Questions	Age Groups				χ2	*p*
	18–19	20–22	23–25	>26		
Lack of symptoms from genital areas?	17 (28.8%)	42 (26.6%)	23 (19.3%)	22 (33.3%)	4.881	0.181
Painful menstruation	18 (30.5%)	60 (38.0%)	34 (28.6%)	18 (27.3%)	3.930	0.269
Intensive periods or bleeding between periods?	25 (42.4%)	77 (48.4%)	45 (37.8%)	45 (68.2%)	16.496	0.001 *
Irregular menstruation or lack of menstruation?	19 (32.2%)	69 (43.7%)	37 (31.1%)	39 (59.1%)	16.130	0.001 *
Smelly vaginal discharge?	20 (33.9%)	74 (46.8%)	44 (37.0%)	45 (68.2%)	20.495	<0.001 *
Blood-stained mucus?	30 (50.8%)	90 (57.0%)	72 (60.5%)	58 (87.9%)	23.731	<0.001 *
Itching in the genital area?	16 (27.1%)	60 (38.0%)	31 (26.1%)	25 (37.9%)	6.012	0.111
Bleeding intercourse?	22 (37.3%)	67 (42.4%)	41 (34.5%)	47 (71.2%)	25.474	<0.001 *
High fever?	13 (22.0%)	71 (44.9%)	40 (33.6%)	40 (60.6%)	22.986	<0.001 *
Is it a test that gives a 100% chance of early diagnosis of cervical cancer?	42 (71.2%)	107 (67.7%)	78 (65.5%)	27 (40.9%)	17.386	0.001 *
Is the test painful?	42 (71.2%)	132 (83.5%)	90 (75.6%)	50 (75.8%)	5.025	0.170
Is the test painful?	53 (89.8%)	151 (95.6%)	114 (95.8%)	63 (95.5%)	3.428	0.330
Is it possible to be tested free of charge?	24 (40.7%)	65 (41.1%)	53 (44.5%)	49 (74.2%)	23.010	<0.001 *
Is it sufficient to do the test only once in order to eliminate the risk of cervical cancer?	48 (81.4%)	145 (91.8%)	111 (93.3%)	58 (87.9%)	7.231	0.065
Can the test cause serious complications?	58 (98.3%)	148 (93.7%)	110 (92.4%)	63 (95.5%)	2.799	0.424
Is it possible for the Pap smear to increase the susceptibility to cervical cancer in the future?	58 (98.3%)	152 (96.2%)	110 (92.4%)	63 (95.5%)	3.638	0.303
How long (year) after sexual initiation should women undergo the test?	32 (54.2%)	103 (65.2%)	78 (65.5%)	53 (80.3%)	14.991	0.091
How often (year) should women do the test?	31 (52.5%)	108 (68.4%)	87 (73.1%)	62 (93.9%)	35.851	<0.001 *

* Values are significant.

## Data Availability

The data are available from the authors on personal request.

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
