# Peer review of "Cervical Cancer Prevention Knowledge (Cckp-64) among Female Students in Novi Sad, Serbia during COVID-19 Pandemic"

_healthcare, 2023, doi:10.3390/healthcare11101400_

Round 1
Reviewer 1 Report
Overall the paper is written well in all aspects. I have few questions and suggestions for the authors.
1) Re-read again whole article and rectify the grammetical mistakes and spelling.
2) In abstract, the result section is very weak. I suggest that add significant results.
3) In introduction, use some studies that shows the importance of pap and vaccine prevention. Further also add some statistics regarding pap smear and vaccines.
4) What is validity of questionnare.
5) Which recoemmended tools you used for development of questions.
6) Is there any intervention is this study?
7) Table 4 , A part is un-related, the questions are not related with prevention. So modify it.
Author Response
Overall the paper is written well in all aspects. I have few questions and suggestions for the authors.
- Re-read again whole article and rectify the grammetical mistakes and spelling.
This rigorous review process has improved the clarity and coherence of the paper, making it more academically sound and polished. With this in mind, we are confident that the revised manuscript will communicate our ideas effectively to the academic community and beyond.
2) In abstract, the result section is very weak. I suggest that add significant results.
Thank you for your feedback regarding the result section of the abstract. It is important to provide a comprehensive overview of the study's findings to ensure that readers can better understand its contributions to the field. We have revised this section and added more results.
- In introduction, use some studies that shows the importance of pap and vaccine prevention. Further also add some statistics regarding pap smear and vaccines.
We are pleased to inform you that the introduction section of the manuscript has been updated with additional information on the preventative measures of PAP smear and vaccination. Furthermore, relevant statistics have been incorporated to support the importance of these measures in disease prevention
4) What is validity of questionnare.
The questionnaire CCKP-64 has already been validated in multiple studies across the globe. The studies are https://www.ncbi.nlm.nih.gov/pmc/articles/PMC3996269/
https://www.ncbi.nlm.nih.gov/pmc/articles/PMC6988189/
5) Which recoemmended tools you used for development of questions.
So, this study was Developed based on the CCKP-64, the questionnaire which consisted of 64 close-ended questions distributed across six domains and was based on adapt guidelines from the European Organization for research and Treatment of Cancer (EORTC) is designed to assess knowledge of cervical cancer prevention among high school and college females students. The CCKP-64 was already validated in multiple studies and with the internal consistency alpha of 0.914 this study indicates a high level of ratability.
6) Is there any intervention is this study?
The study didn’t include any intervention. That will be the objective of a followed up project and study. The purpose of this study was to examine the level of knowledge about cervical cancer prevention among students at the University of Novi Sad in Serbia during the COVID-19 pandemic. The results of this survey will give us a quick overview of the current level of understanding about cervical cancer prevention, help us to stay informed about the current state of affairs, and also draw the attention of government bodies to the importance of educating female students about preventing cervical cancer.
7) Table 4 , A part is un-related, the questions are not related with prevention. So modify it.
This part is an essential part of the CCKP-64 and that’s why we added it to the research. Some research papers also prove a limited contribution to the cervical cancer development due to these factors.
Reviewer 2 Report
The manuscript "Cervical Cancer Prevention Knowledge (Cckp-64) among Female Students in Novi Sad, Serbia during COVID-19 Pandemic" is an interesting manuscript on the knowledge of cervical cancer prevention among female students at the University of Novi Sad during the COVID-19 pandemia. The work is not so original, but it is well-performed and structured. The design of the project is appropriate and the results are significant. The statistical analysis is well conducted and the language is acceptable. The main concern of this work is related to the role of Covid-19 in the knowledge of these students: could the COVID-19 pandemic have reduced the opportunity for these students to prevent cervical cancer? and could the COVID-19 pandemic have reduced the opportunity for these students to have an improved knowledge of cervical cancer? Considering that it is specified also in the title of the manuscript, it is important to define which is the role of COVID-19 in this context. Otherwise, the work is well structured, giving important information to scientific literature and new perspectives for future studies.
Author Response
The main concern of this work is related to the role of Covid-19 in the knowledge of these students: could the COVID-19 pandemic have reduced the opportunity for these students to prevent cervical cancer? and could the COVID-19 pandemic have reduced the opportunity for these students to have an improved knowledge of cervical cancer? Considering that it is specified also in the title of the manuscript, it is important to define which is the role of COVID-19 in this context. Otherwise, the work is well structured, giving important information to scientific literature and new perspectives for future
The COVID-19 pandemic has disrupted healthcare services worldwide, leading to significant changes in medical practice, including the delivery of routine health services. Screening and vaccination programs for cervical cancer have been adversely impacted by the pandemic, leading to concerns about potential delays in diagnosis and treatment of the disease. As such, it is crucial to assess the level of knowledge of cervical cancer prevention among the population during the pandemic to address any gaps and improve public health outcomes.
Our study sought to evaluate the level of knowledge of cervical cancer prevention among female students at the University of Novi Sad during the COVID-19 pandemic. We used the Cervical Cancer Knowledge Prevention-64 (CCKP-64) tool to assess the participants' knowledge of cervical cancer risk factors, screening, prevention, and early detection.
Our findings highlighted the need for educational interventions to improve knowledge of cervical cancer prevention, particularly during the pandemic. We identified gaps in knowledge related to cervical cancer risk factors and the importance of regular screening, indicating a need for educational interventions targeted at improving awareness of the disease's early detection and prevention.
We have added this concern in the manuscript.
Reviewer 3 Report
Dear Authors
Thank you for the opportunity to read your manuscript, which I read with great interest.
The manuscript is well structured, however, it needs some changes that will improve it significantly. Below you will find some points in the manuscript that need clarification, refinement, reanalysis, rewriting, and/or additional information and suggestions on what can be done to improve it.
Title - Appropriate
Abstract - Some descriptors should be revised and fit with DeCS/Mesh:
Maternal gynecologic cancer - correct to Genital Neoplasms, Female;
gynecologic disease - correct to Genital Diseases, Female;
medical disease - correct to Disease
The rest should start with capital letters.
Section 1 (Introduction) - this section needs some adjustments, as some information and/or points are missing or unclear, and should be included or clarified, I will present some items:
o What is the importance of doing this research/contribution it brings to the literature in the field?
o Why should readers be interested?
What problem/issue does this research solve/fill?
o How will the proposed study address this deficiency/lacuna/problem and provide a unique contribution to the literature.
o The purpose of the study should be in line with what is presented in the abstract.
Section 2 (Materials and Methods) - in this section some points should be clarified and improved, namely:
- The nature of the study and the study design.
- Clarify the type of sampling, the number of variables and what are the variables under study.
- The recruitment procedures of the women, also leave me in doubt, as well as the way of collecting the information.
- The recruitment procedures of the participants.
Section 3 (Results) - This section leaves me with doubts, as some methodological information is missing, as mentioned earlier, I am unable to clearly evaluate the data.
- Review line 142 the information "102 (25.4%)"
- Review formatting of tables
Section 4 (Discussion) - the discussion and conclusion should be separate. The discussion should be improved and supported by more authors.
Section 5 (Conclusion) - does not present this section, should be presented, and could present strategies to reduce the problem under study.
Author Response
Thank you for the opportunity to read your manuscript, which I read with great interest.
Title – Appropriate
Thank You very much.
Abstract - Some descriptors should be revised and fit with DeCS/Mesh:
Maternal gynecologic cancer - correct to Genital Neoplasms, Female;
gynecologic disease - correct to Genital Diseases, Female;
medical disease - correct to Disease
The rest should start with capital letters.
The aformentioned comments have been adopted and added to the manuscript.
Section 1 (Introduction) - this section needs some adjustments, as some information and/or points are missing or unclear, and should be included or clarified, I will present some items:
o What is the importance of doing this research/contribution it brings to the literature in the field?
We have added this point “This research makes a significant contribution to the literature in the field of cervical cancer prevention, particularly during the COVID-19 pandemic. By investigating the level of knowledge about cervical cancer prevention among students, our study provides insight into the effectiveness of current educational programs on cervical cancer prevention and identifies areas for improvement. Our findings can inform the development of new educational strategies and interventions to increase awareness and understanding of cervical cancer prevention.”
o Why should readers be interested?
Readers should be interested in this study because it addresses a critical public health issue, and the results could inform policies and programs aimed at reducing the burden of cervical cancer in the population. As the COVID-19 pandemic has disrupted healthcare services globally, this research provides timely information on the impact of the pandemic on cervical cancer prevention efforts
What problem/issue does this research solve/fill?
We have added this point in the introcution “This research makes a significant contribution to the literature in the field of cervical cancer prevention, particularly during the COVID-19 pandemic”
o How will the proposed study address this deficiency/lacuna/problem and provide a unique contribution to the literature.
We have added this point in the introduction. “The proposed study addresses the deficiency in the literature on the level of knowledge about cervical cancer prevention during the COVID-19 pandemic. The pandemic has disrupted healthcare services, including cervical cancer screening and vaccination programs, making it crucial to assess the level of knowledge about cervical cancer prevention among the population.”
o The purpose of the study should be in line with what is presented in the abstract.
We have corrected this line “The objective of our study was to evaluate the knowledge of cervical cancer prevention among female students at the University of Novi Sad during the COVID-19 pandemic using the Cervical Cancer Knowledge Prevention-64 (CCKP-64) tool and to address the deficiency in the literature on the level of knowledge about cervical cancer prevention during the COVID-19 pandemic.”
Section 2 (Materials and Methods) - in this section some points should be clarified and improved, namely:
- The nature of the study and the study design.
This cross-sectional study was performed in the period of the academic 2021/2022 year among the students University of Novi Sad in Serbia during the COVID-19 pandemic, so the design was cross-sectional.
- Clarify the type of sampling, the number of variables and what are the variables under study.
We have clarified the aformentioned comments: The collection of official email addresses assigned to students by the University of Novi Sad was achieved through their information system, while the selection of participants for the study was carried out using convenience sampling.; Developed based on the CCKP-64 [12], the questionnaire which consisted of 64 close-ended questions and 67 variables distributed across six domains and was based on adapt guidelines from the European Organization for research and Treatment of Cancer (EORTC) is designed to assess knowledge of cervical cancer prevention among high school and college females students.”
- The recruitment procedures of the women, also leave me in doubt, as well as the way of collecting the information.
All female students were recruted with respect of the exclusion criteria, diue to the fact that Medical science students are aware about the mentioned topic leading to a potentail bias and confounding.
- The recruitment procedures of the participants.
It is added to the text : The study was approved by the Ethical Committee of the University of Novi Sad (01-39/202/1) with the exception of the field of medical sciences. Students completed an anonymous online questionnaire regarding Cervical Cancer Knowledge Prevention-64 (CCKP-64) using Google Forms Administration App which prevented multiple responses per e-mail. The collection of official email addresses assigned to students by the University of Novi Sad was achieved through their information system, while the selection of participants for the study was carried out using convenience sampling
Section 3 (Results) - This section leaves me with doubts, as some methodological information is missing, as mentioned earlier, I am unable to clearly evaluate the data.
- Review line 142 the information "102 (25.4%)"
We have reviewed the ine 142.
- Review formatting of tables
Table formatting has been corrected.
Section 4 (Discussion) - the discussion and conclusion should be separate. The discussion should be improved and supported by more authors.
We have corrected accoridng to the comments.
Section 5 (Conclusion) - does not present this section, should be presented, and could present strategies to reduce the problem under study.
This section has been added.
Round 2
Reviewer 3 Report
Dear Authors
Thank you for the changes made, which improve the quality of the article.
I call your attention again to the fact that the objective in the abstract must be identical to the one presented in the introduction and vice-versa, because currently I see that in the introduction there are two objectives mentioned and in the abstract there is only one objective.
The tables have been improved, but should follow the standards of the journal.
Author Response
Dear Reviewer,
we have corrected all mentioned issues.
SIncerely